# Substantially Altered Expression Profile of Diabetes/Cardiovascular/Cerebrovascular Disease Associated microRNAs in Children Descending from Pregnancy Complicated by Gestational Diabetes Mellitus—One of Several Possible Reasons for an Increased Cardiovascular Risk

**DOI:** 10.3390/cells9061557

**Published:** 2020-06-26

**Authors:** Ilona Hromadnikova, Katerina Kotlabova, Lenka Dvorakova, Ladislav Krofta, Jan Sirc

**Affiliations:** 1Department of Molecular Biology and Cell Pathology, Third Faculty of Medicine, Charles University, 10000 Prague, Czech Republic; katerina.kotlabova@lf3.cuni.cz (K.K.); lenka.dvorakova@lf3.cuni.cz (L.D.); 2Institute for the Care of the Mother and Child, Third Faculty of Medicine, Charles University, 14700 Prague, Czech Republic; ladislav.krofta@upmd.eu (L.K.); jan.sirc@upmd.eu (J.S.)

**Keywords:** BMI, bioinformatics, cardiovascular risk, children, echocardiography, gestational diabetes mellitus, microRNA expression, miRWalk2.0 database, prehypertension/hypertension, screening

## Abstract

Gestational diabetes mellitus (GDM), one of the major pregnancy-related complications, characterized as a transitory form of diabetes induced by insulin resistance accompanied by a low/absent pancreatic beta-cell compensatory adaptation to the increased insulin demand, causes the acute, long-term, and transgenerational health complications. The aim of the study was to assess if alterations in gene expression of microRNAs associated with diabetes/cardiovascular/cerebrovascular diseases are present in whole peripheral blood of children aged 3–11 years descending from GDM complicated pregnancies. A substantially altered microRNA expression profile was found in children descending from GDM complicated pregnancies. Almost all microRNAs with the exception of miR-92a-3p, miR-155-5p, and miR-210-3p were upregulated. The microRNA expression profile also differed between children after normal and GDM complicated pregnancies in relation to the presence of overweight/obesity, prehypertension/hypertension, and/or valve problems and heart defects. Always, screening based on the combination of microRNAs was superior over using individual microRNAs, since at 10.0% false positive rate it was able to identify a large proportion of children with an aberrant microRNA expression profile (88.14% regardless of clinical findings, 75.41% with normal clinical findings, and 96.49% with abnormal clinical findings). In addition, the higher incidence of valve problems and heart defects was found in children with a prior exposure to GDM. The extensive file of predicted targets of all microRNAs aberrantly expressed in children descending from GDM complicated pregnancies indicates that a large group of these genes is involved in ontologies of diabetes/cardiovascular/cerebrovascular diseases. In general, children with a prior exposure to GDM are at higher risk of later development of diabetes mellitus and cardiovascular/cerebrovascular diseases, and would benefit from dispensarisation as well as implementation of primary prevention strategies.

## 1. Introduction

Gestational diabetes mellitus (GDM) represents a pregnancy-related complication with the onset during the second or third trimester of gestation with worldwide increasing prevalence ranging from 7 to 14% [1,2]. GDM is defined as a transitory form of diabetes induced by insulin resistance accompanied by a low/absent pancreatic beta-cell compensatory adaptation to the increased insulin demand. Pancreatic beta-cell dysfunction is just a central component of the pathogenesis of GDM [1,2,3,4,5,6].

MicroRNAs are important metabolic and developmental regulators during gestation that play a role in the onset of GDM [7]. Recent studies have evaluated the expression of microRNAs playing a role in glucose homeostasis, insulin sensitivity, and beta-cell function in different sample types (placenta, umbilical vein endothelial cells, whole blood, plasma, and serum) with the aim to assess their potential as diagnostic or prognostic biomarkers of GDM [7,8,9,10,11,12,13,14,15,16,17,18,19,20,21].

Accumulating data suggest that exposure to hyperglycemia in utero, as occurs in gestational diabetes mellitus, may expose offspring to short-term and long-term adverse effects [22,23,24,25].

Young offspring aged 3–5 years of GDM complicated pregnancies were reported to have increased BMI, skinfold thickness, body fat, blood pressure, altered lipid profiles, and glucose metabolism [26]. Several consecutive studies confirmed higher mean values of systolic blood pressure and higher prevalence of hypertension in young offspring (at the age of 3–6 years) of mothers with GDM compared with their counterparts born to mothers with a normal course of gestation [27,28]. An independent association between the occurrence of maternal GDM and abnormal glucose tolerance, obesity, and higher BP was confirmed in children at seven years of age [29]. Another study indicated that also children at the age of 10–14 years had an impaired glucose tolerance as a consequence of in utero exposition to maternal GDM [30].

Parallel, in 17-year-old offspring of mothers with GDM and higher mean BMI, systolic and diastolic BP values were observed when compared to no-recorded-GDM offspring [31]. Nevertheless, another recent study performed on youth (at the age of 10.6–16.9 years) demonstrated that maternal GDM was related to altered lipid profile and higher systolic blood pressure only in a sex-specific manner [32]. A higher total and LDL cholesterol were detected only in girls and a higher systolic blood pressure only in boys previously exposed in utero to GDM [32]. Even a significant association between maternal GDM and the rate of hospitalization for subsequent cardiovascular related morbidities (pre-defined set of ICD-9 codes) was noted in an 18-year follow-up study [33].

Another population-based cohort study with a 40-year follow-up revealed the GDM association with an increased occurrence of cardiovascular diseases in early adulthood. Varied increased rates of specific early onset cardiovascular diseases, particularly heart failure, hypertensive disease, deep vein thrombosis, and pulmonary embolism, were observed in the offspring previously affected with GDM [34]. Similarly, young adults descending from pregnancies with an impaired glucose tolerance were at an increased risk of development of type 2 diabetes [23,35,36]. The cumulative risk of type 2 diabetes reached 15% in the offspring at the age of 20 years, and about 30% at the age of 24 years [23,35].

Furthermore, children with in utero exposition to GDM were demonstrated to have an impaired neurophysiology (reduced cortical excitability, neuroplasticity, and salivary cortisol) [37] and to be at an increased risk of neurodevelopmental difficulties, including attention deficit hyperactivity disorder [38,39], autism spectrum disorders [40,41], impaired motor development [42], and neuropsychiatric disorders [43].

In addition, children descending from GDM complicated pregnancies were also reported to have an increased risk of paediatric ophthalmic morbidity [44]. Moreover, a small risk of childhood asthma was observed after exposure to GDM requiring medication only [45].

In general, the aim of this follow-up study was to assess if alterations in the gene expression of microRNAs are present in children descending from GDM complicated pregnancies.

In detail, the expression profile of microRNAs associated with diabetes/cardiovascular/cerebrovascular diseases was assessed in whole peripheral venous blood (white blood cells) of children aged 3 to 11 years prenatally exposed to GDM with the goal to assess to what extent fetal and environmental programming predisposed the affected children to the development of diabetes mellitus, cardiovascular, and cerebrovascular diseases.

The hypothesis of the assessment of potential diabetes/cardiovascular risk in children prenatally exposed to GDM was founded on the fact that particular microRNAs play a key role in the inducement and progress of diabetes mellitus and cardiovascular/cerebrovascular diseases (Table 1) [46,47,48,49,50,51,52,53,54,55,56,57,58,59,60,61,62,63,64,65,66,67,68,69,70,71,72,73,74,75,76,77,78,79,80,81,82,83,84,85,86,87,88,89,90,91,92,93,94,95,96,97,98,99,100,101,102,103,104,105,106,107,108,109,110,111,112,113,114,115,116,117,118,119,120,121,122,123,124,125,126,127,128,129,130,131,132,133,134,135,136,137,138,139,140,141,142,143,144,145,146,147,148,149,150,151,152,153,154,155,156,157,158,159,160,161,162,163,164,165,166,167,168,169,170,171,172,173,174,175,176,177,178,179,180,181,182,183,184,185,186,187,188,189,190,191,192,193,194,195,196,197,198,199,200,201,202,203,204,205,206,207,208,209,210,211,212,213,214,215,216,217,218,219,220,221].

Previously, by searching the Medline database we identified a large scale of microRNAs playing a role in pathogenesis of diabetes mellitus and cardiovascular/cerebrovascular diseases. Finally, we selected for the study a shortlist of 29 microRNAs demonstrated repetitively by numerous scientific teams to be associated with normal stages (development and homeostasis of the cardiovascular system, angiogenesis, and adipogenesis) and pathological conditions and diseases (vascular endothelial dysfunction and inflammation, hypoxia, hypertension and regulation of hypertension-related genes, obesity, dyslipidaemia, atherosclerosis and atherosclerotic plaque formation, insulin resistance, diabetes mellitus and diabetes-related complications, metabolic syndrome, cardiovascular diseases involving the blood vessels (coronary and peripheral artery diseases, carotid artery disease, pulmonary arterial hypertension, cerebrovascular disease, aortic and intracranial aneurysms), cardiovascular diseases involving the heart (congenital heart disease, cardiomyopathies, cardiac dysrhythmias, hypertensive heart disease, myocardial disease, valvular heart disease, inflammatory heart disease, rheumatic heart disease, pulmonary heart disease, and heart failure), chronic kidney disease, ischemia/reperfusion injury, cardiac regeneration and cachexia) (Table 1) [46,47,48,49,50,51,52,53,54,55,56,57,58,59,60,61,62,63,64,65,66,67,68,69,70,71,72,73,74,75,76,77,78,79,80,81,82,83,84,85,86,87,88,89,90,91,92,93,94,95,96,97,98,99,100,101,102,103,104,105,106,107,108,109,110,111,112,113,114,115,116,117,118,119,120,121,122,123,124,125,126,127,128,129,130,131,132,133,134,135,136,137,138,139,140,141,142,143,144,145,146,147,148,149,150,151,152,153,154,155,156,157,158,159,160,161,162,163,164,165,166,167,168,169,170,171,172,173,174,175,176,177,178,179,180,181,182,183,184,185,186,187,188,189,190,191,192,193,194,195,196,197,198,199,200,201,202,203,204,205,206,207,208,209,210,211,212,213,214,215,216,217,218,219,220,221].

Furthermore, the MiRWalk database (http://www.umm.uni-heidelberg.de/apps/zmf/mirwalk/) was utilized to acquire data on the dysregulation of microRNAs, in particular, the abovementioned human disease and phenotype ontologies and OMIM disorders.

To our knowledge, no data on expression profiles of microRNAs associated with diabetes mellitus, cardiovascular, and cerebrovascular diseases in whole peripheral venous blood (white blood cells) of children descending from GDM affected pregnancies are currently available. The only study which was reported in this field was dedicated to profiling of mir-15 family members, affecting insulin signalling pathway, in skeletal muscle biopsies of adult offspring of women with a history of GDM [222].

## 2. Materials and Methods

### 2.1. Participants

The study had a prospective design, that ran from August 2016 to October 2019, and included Caucasian children aged 3 to 11 years descending from normal (*n* = 85) and GDM complicated pregnancies (*n* = 118). Children of equal age descending from pregnancies with a normal course of gestation were chosen as a control group.

Ninety-eight children were exposed to a GDM pregnancy on diet only, and 20 children descended from a GDM complicated pregnancy on the combination of diet and therapy (in nineteen pregnancies insulin was administrated and in one pregnancy metformin was prescribed). According to the Institutional Review Board (IRB) Guidelines for gestational diabetes mellitus mothers were divided into groups based on preconception BMI (below 18.5, 18.5–24.9, 25.0–29.9, above 30.0), the achievement of optimal total weight gain during pregnancy (12.5–18.0, 11.5–16.0, 7.0–11.5, 5.0–9.0 kg), and the achievement of optimal week weight gain during the second and the third trimesters of gestation (0.5–0.6, 0.4–0.5, 0.2–0.3, 0.2–0.3 kg) were monitored.

The clinical data of children descending from normal and GDM complicated pregnancies are summarized in Table 2. The preconceptional and gestational clinical data of mothers with normal and GDM complicated pregnancies are involved as well.

Children descending from normal pregnancies were healthy infants born after 37 completed weeks of gestation with the weight >2500 g. Normal pregnancies were those ones with the absence of medical, obstetrical, or surgical complications.

Gestational diabetes mellitus was diagnosed if any degree of glucose intolerance appeared for the first-time during gestation [223,224,225]. Concerning the diagnosis and classification of hyperglycemia in pregnancy the guidelines of The International Association of Diabetes and Pregnancy Study Groups (IADPSG) were followed [223]. The first screening was performed during the first trimester of gestation with the aim to detect women with overt diabetes (fasting plasma glucose level is ≥7.0 mmol/L) and women with GDM (fasting plasma glucose level ≥5.1–<7.0 mmol/L). The second screening phase, 2 h 75-g OGTT, was performed during 24–28 weeks of gestation only in those women not previously diagnosed to have overt diabetes or GDM. The second screening phase identified GDM if the fasting plasma glucose level was ≥5.1 mmol/L, or the 1 h plasma glucose was ≥10.0 mmol/L, or the 2-h plasma glucose was ≥8.5 mmol/L [223].

Children with inborn defects or chromosomal disorders, as well as children descending from pregnancies demonstrating other complications were excluded from the inclusion into the study.

Informed consent was obtained from all participants included in the study. Two independent Ethics Committees (one at the Institute for the Care of the Mother and Child, and the second one at the Third Faculty of Medicine, Charles University) approved the study (grant no. AZV 16-27761A, long-term monitoring of complex cardiovascular profile in the mother, foetus, and offspring descending from pregnancy-related complications, dates of approval: 27 March, 2014 and 28 May, 2015). All procedures were also in compliance with the Helsinki Declaration of 1975, as revised in 2000.

### 2.2. BP and Echocardiography Measurements, BMI Assessment

Standardized BP and echocardiography measurements, and BMI assessment were performed as previously described [226]. In brief, the average of the last two systolic blood pressure (SBP) and diastolic blood pressure (DBP) values was taken under consideration for data analyses. A normal BP was defined as SBP and DBP that was below the 90th percentile for gender, age, and height. Hypertension was defined as an average SBP or DBP ≥95th percentile on at least three separate occasions. Prehypertension was defined as an average SBP or DBP within the range of ≥90th percentile and <95th percentile [227].

The age- and sex-specific BMI Percentile Calculator was used to calculate BMI in children (https://www.cdc.gov/healthyweight/assessing/bmi/childrens_bmi/about_childrens_bmi.html). BMI above the 5th percentile and below the 85th percentile was considered as normal BMI. Children with BMI above the 85th percentile and below the 95th percentile were in the overweight category. Children with BMI equal to or greater than the 95th percentile were in the obese category.

A complete two-dimensional echocardiography was performed using the Philips HD15 ultrasound machine (Philips Ultrasound, Bothell, WA, USA) with a sector array transducer (3–8 MHz) incorporating colour flow, pulse wave, and continuous wave Doppler measurements with an adaptive technology. A complete two-dimensional echocardiography was performed by an investigator experienced with paediatric echocardiography. Children with abnormal findings were referred to a paediatric cardiologist.

### 2.3. Processing of Samples

Samples of unclotted whole peripheral venous blood (200 µL) were processed as previously described [226]. Briefly, homogenized cell lysates were prepared as soon as possible after blood collection using a QIAamp RNA Blood Mini Kit (Qiagen, Hilden, Germany, no. 52304).

Afterwards, total RNA was extracted using a mirVana microRNA Isolation kit (Ambion, Austin, TX, USA, no. AM1560). The isolated RNA was treated with DNase I (Thermo Fisher Scientific, Carlsbad, CA, USA, no. EN0521).

### 2.4. Reverse Transcription

Particular microRNAs were transcribed into cDNA using microRNA-specific stem-loop RT primers, components of TaqMan MicroRNA Assays and TaqMan MicroRNA Reverse Transcription Kit (Applied Biosystems, Branchburg, NJ, USA, no. 4366597) as previously described [226]. A total reaction volume of the reaction was 10 µL. The reverse transcriptase reaction was performed following the guidelines for a 7500 Real-Time PCR system (Applied Biosystems, Branchburg, NJ, USA): 30 min at 16 °C, 30 min at 42 °C, 5 min at 85 °C, and then held at 4 °C.

### 2.5. Relative Quantification of microRNAs

Relative quantification of microRNAs by real-time PCR was performed as previously described [226]. cDNA (3 µL) was mixed with specific TaqMan MGB primers and probes (TaqMan MicroRNA Assay, Applied Biosystems, Branchburg, NJ, USA), and the constituents of the TaqMan Universal PCR Master Mix (Applied Biosystems, Branchburg, NJ, USA, no: 4318157) in a total reaction volume of 15 µL. The samples were regarded as positive if Ct (threshold cycle) was below 40 (Ct < 40).

The comparative Ct method was used to determine the expression of each microRNA [228]. The expression of studied microRNAs was normalized to a geometric mean of RNU58A and RNU38B, endogenous controls with the lowest expression variability between studied samples [229]. A reference sample was used throughout the study for relative quantification. As a reference sample we used small RNAs enriched RNA fraction extracted from the fetal part of one randomly selected placenta of a normally ongoing pregnancy.

### 2.6. Statistical Analysis

Logistic regression was used to compare the presence of abnormal clinical findings (BMI in the category overweight or obesity and/or systolic or diastolic BP values in the category prehypertension or hypertension and/or the presence of valve problems and heart defects) among various groups of children.

The Shapiro-Wilk test was used to assess the data normality [230]. Our experimental data did not follow a normal distribution. Therefore, microRNA levels were compared among the groups of children using the Kruskal-Wallis one-way analysis of variance with a post-hoc test (K-W) for the comparison between multiple groups.

Receivers operating characteristic (ROC) curves were used to assess the areas under the curves (AUC), the optimal cut-off points, and the respective sensitivities at 10.0% false positive rate (FPR) for particular microRNAs (MedCalc Software bvba, Ostend, Belgium). The significance level was established at a *p*-value of *p* < 0.05.

To identify the optimal combinations of microRNA biomarkers logistic regression combined with the ROC curve analysis was used (MedCalc Software bvba, Ostend, Belgium). In brief, in this setting, the power of the model’s predicted values to discriminate between positive and negative cases is quantified by the area under the ROC curve. To perform a full ROC curve analysis the predicted probabilities are first saved and next used as a new variable in the ROC curve analysis. The dependent variable used in logistic regression then acts as the classification variable in the ROC curve analysis dialog box.

Statistica software (version 9.0; StatSoft, Inc., Tulsa, OK, USA) was used to generate box plots of log-normalized gene expression values (RT-qPCR expression, log_10_ 2^−ΔΔCt^) for particular microRNAs. The box plots display the medians, the 75th and 25th percentiles (the upper and lower limits of the boxes), the maximum and minimum values (the upper and lower whiskers), outliers (circles), and extremes (asterisks). Dot plots, all observations are also displayed in the charts.

### 2.7. Bioinformatics Analysis—microRNA-Target Interactions on Disease Ontologies, Human Phenotype Ontologies, and OMIM Disorders

MiRWalk database (http://www.umm.uni-heidelberg.de/apps/zmf/mirwalk/) and the Predicted Target module [231] were utilized to acquire data on predicted targets for microRNAs that have been identified to be dysregulated in whole peripheral venous blood of children descending from GDM complicated pregnancies.

MiRWalk is a comprehensive database that provides in addition to other things also information on human microRNA predicted and/or validated target genes, information on microRNA-target interactions on disease ontologies, human phenotype ontologies and OMIM disorders, and information on possible interactions between microRNAs and genes associated with KEGG, Panther and Wiki pathways.

Only those predicted targets involved in ontologies of human diseases (obesity, hypertension, glucose intolerance, lipid metabolism disease, type 2 diabetes mellitus, heart septal defects and heart valve disease, heart disease, heart failure, venous insufficiency, and pulmonary embolism) reported by previous population-based cohort studies in children and young adults descending from GDM complicated pregnancies [26,27,28,29,30,31,32,33,34,35,36] are presented below.

### 2.8. Workflow of the Study

Figure 1 presents the workflow of the study including the particular groups of the participants, the performance of relative quantification of microRNAs and bioinformatics analyses, and the variety of statistical analyses applied.

## 3. Results

### 3.1. Higher Incidence of Valve Problems and Heart Defects in a group of Children Descending from GDM Complicated Pregnancies

We distributed children descending from normal and GDM complicated pregnancies into groups based on the results of anamnesis and the results of consecutive clinical examination. The group of children with abnormal clinical findings consisted of those ones who were already dispensarised in the department of paediatric cardiology, or those ones who were overweight/obese, had prehypertension/hypertension, and/or abnormal echocardiogram findings during the visit (in total: FG, *n* = 35; GDM, *n* = 57).

In detail, the studied groups of children with abnormal clinical findings consisted of those ones already dispensarised in the department of paediatric cardiology (FG, *n* = 8/85; GDM, *n* = 5/118), those ones indicated by the sonographer during the visit to have valve problems and heart defects (FG, *n* = 17/85; GDM, *n* = 42/118) (tricuspid valve regurgitation (FG, *n* = 8/85; GDM, *n* = 23/118), mitral valve regurgitation (FG, *n* = 1/85; GDM, *n* = 1/118), pulmonary valve regurgitation (FG, *n* = 2/85; GDM, *n* = 8/118), bicuspid aortic valve regurgitation (FG, *n* = 1/85; GDM, *n* = 0/118), ventricular septum defect (FG, *n* = 1/85; GDM, *n* = 0/118), atrial septum defect (FG, *n* = 1/85; GDM, *n* = 2/118), foramen ovale apertum (FG, *n* = 5/85; GDM, *n* = 11/118), arrhythmia (FG, *n* = 1/85; GDM, *n* = 1/118)], those ones confirmed over several visits to have a high BP (FG, *n* = 15/85; GDM, *n* = 19/118) (SBP and/or DBP ≥ 90th percentile evaluated by the Age-based Paediatric Blood Pressure Reference Charts calculator) and/or high BMI (FG, *n* = 8/85; GDM, *n* = 6/118) (BMI >85th percentile evaluated by the BMI Percentile Calculator for Child and Teens).

The group of children with normal clinical findings consisted of children with normal anamnesis, normal BP, normal BMI, and normal reference values of echocardiographic measurements (FG, *n* = 50/85; GDM, *n* = 61/118).

Logistic regression revealed no difference in the incidence of overweight/obesity and/or prehypertension/hypertension between the groups of children descending from normal and GDM complicated pregnancies. Nevertheless, a higher incidence of valve problems and heart defects was observed in a group of children descending from GDM complicated pregnancies (Table 3).

### 3.2. Only Just miR-21-5p Indicates a Trend Towards Differentiation between Children with Normal and Abnormal Clinical Findings (BMI, Blood Pressure, and Echocardiogram Findings) Descending from Normal Pregnancies

Furthermore, we compared the microRNA expression profile between individual groups of children with respect to clinical findings. From the ROC curve analysis, it is apparent that only miR-21-5p (a sensitivity of 17.14% at a specificity of 90.0%) trended to differentiate between children descending from normal pregnancies with normal and abnormal clinical findings (BMI, blood pressure, and echocardiogram findings) (Appendix A).

### 3.3. Only Just miR-29a-3p and miR-92a-3p Indicate a Slight Differentiation Between Children with Normal and Abnormal Clinical Findings (BMI, Blood Pressure, and Echocardiogram Findings) Descending from GDM Complicated Pregnancies

The performance of the ROC curve analysis revealed that miR-29a-3p (12.28%) and miR-92a-3p (12.28%) were able to slightly differentiate at 10.0% FPR within children descending from GDM complicated pregnancies based on the presence of normal and abnormal clinical findings (BMI, blood pressure, and echocardiogram findings) (Appendix A).

### 3.4. Substantially Altered Expression Profile of Diabetes/Cardiovascular/Cerebrovascular Disease Associated microRNAs in Children Descending from Pregnancy Complicated by Gestational Diabetes Mellitus

With regard to the assessment of the effect of maternal pregnancy complication on postnatal microRNA expression profile, we further compared microRNA gene expression between children descending from normal and GDM complicated pregnancies irrespective of the clinical findings (overweight/obesity, prehypertension/hypertension, and/or valve problems and heart defects). The K-W and subsequent ROC curve analysis revealed a significant dysregulation of multiple microRNAs in children descending from GDM complicated pregnancies (Figure 2, Appendix A).

The sensitivity at 10.0% FPR for miR-1-3p (40.68%), miR-16-5p (14.41%), miR-17-5p (22.88%), miR-20a-5p (27.97%), miR-20b-5p (29.66%), miR-21-5p (16.10%), miR-26a-5p (17.80%), miR-29a-3p (19.49%), miR-92a-3p (6.78%), miR-103a-3p (20.34%), miR-125b-5p (24.58%), miR-126-3p (25.42%), miR-133a-3p (22.03%), miR-143-3p (18.64%), miR-146a-5p (8.47%), miR-155-5p (5.08%), miR-181a-5p (16.10%), miR-195-5p (27.97%), miR-199a-5p (11.02%), miR-210-3p (15.25%), miR-221-3p (17.80%), miR-499a-5p (27.97%), and miR-574-3p (18.64%) was observed (Figure 2, Appendix A).

MicroRNAs with a poor sensitivity at 10.0% FPR (miR-92a-3p, miR-146a-5p, miR-155-5p, and miR-199a-5p) were not further used for diabetes mellitus and cardiovascular risk assessment in children descending from GDM complicated pregnancies.

Screening based on a combination of microRNAs with a good sensitivity (miR-1-3p, miR-16-5p, miR-17-5p, miR-20a-5p, miR-20b-5p, miR-21-5p, miR-26a-5p, miR-29a-3p, miR-103a-3p, miR-125b-5p, miR-126-3p, miR-133a-3p, miR-143-3p, miR-181a-5p, miR-195-5p, miR-210-3p, miR-221-3p, miR-499a-5p, and miR-574-3p) showed the highest accuracy for children prenatally exposed to GDM (AUC 0.965, *p* < 0.001, sensitivity 93.22%, specificity 87.06%, cut off >0,477932153). At 10.0% FPR, it was able to identify 88.14% children with an increased risk of later development of diabetes and/or cardiovascular/cerebrovascular diseases (Figure 2).

### 3.5. Dysregulation of Multiple microRNAs in Children Descending from GDM Complicated Pregnancies with Normal Clinical Findings (Normal BMI, Blood Pressure, and Echocardiogram Findings)

Furthermore, we compared the microRNA expression profile between individual groups of children with respect to clinical findings. Despite the presence of normal clinical findings in both groups, that were compared, the performance of the ROC curve analysis revealed that miR-1-3p (57.38%), miR-16-5p (9.84%), miR-17-5p (31.15%), miR-20a-5p (29.51%), miR-20b-5p (32.79%), miR-21-5p (21.31%), miR-23a-3p (26.23%), miR-26a-5p (8.20%), miR-29a-3p (22.95%), miR-100-5p (22.95%), miR-103a-3p (22.95%), miR-125b-5p (36.07%), miR-126-3p (24.59%), miR-133a-3p (27.87%), miR-143-3p (16.39%), miR-146a-5p (9.84%), miR-181a-5p (21.31%), miR-195-5p (19.67%), miR-210-3p (13.11%), miR-221-3p (18.03%), miR-499a-5p (29.51%), and miR-574-3p (16.39%) differentiated between children descending from GDM complicated and normal pregnancies with a various sensitivity at a specificity of 90.0% (Figure 3, Appendix A).

Combined screening of miR-1-3p, miR-17-5p, miR-20a-5p, miR-20b-5p, miR-21-5p, miR-23a-3p, miR-29a-3p, miR-100-5p, miR-103a-3p, miR-125b-5p, miR-126-3p, miR-133a-3p, miR-143-3p, miR-181a-5p, miR-195-5p, miR-221-3p, miR-499a-5p, and miR-574-3p was superior over using individual microRNAs in the assessment of risk of later development of diabetes mellitus and cardiovascular/cerebrovascular diseases (AUC 0.905, *p* < 0.001, sensitivity 88.52%, specificity 82.0%, cut off >0.391945473) in a group of children descending from GDM complicated pregnancies with normal clinical findings. At 10.0% FPR, it was able to identify 75.41% children with an increased diabetic/cardiovascular risk (Figure 3).

### 3.6. Dysregulation of Multiple microRNAs in Children Descending from GDM Complicated Pregnancies with Abnormal Clinical Findings (Abnormal BMI, Blood Pressure, and Echocardiogram Findings)

In addition, it was observed that a set of microRNAs differentiated between the groups of children affected with GDM with abnormal clinical findings (abnormal BMI, blood pressure, and echocardiogram findings) and the controls, children descending from normal pregnancies with normal clinical findings. The sensitivity of individual microRNAs at 10.0% FPR was the following: miR-1-3p (50.88%), miR-17-5p (22.81%), miR-20a-5p (22.81%), miR-20b-5p (26.32%), miR-21-5p (22.81%), miR-29a-3p (15.79%), miR-92a-3p (19.30%), miR-103a-3p (17.54%), miR-126-3p (22.81%), miR-133a-3p (19.30%), miR-143-3p (15.79%), miR-155-5p (7.02%), miR-181a-5p (19.30%), miR-195-5p (29.82%), miR-210-3p (12.28%), miR-221-3p (15.79%), and miR-499a-5p (29.82%) (Figure 4, Appendix A).

Screening based on the combination of microRNAs with a good sensitivity (miR-1-3p, miR-17-5p, miR-20a-5p, miR-20b-5p, miR-21-5p, miR-29a-3p, miR-92a-3p, miR-103a-3p, miR-126-3p, miR-133a-3p, miR-143-3p, miR-181a-5p, miR-195-5p, miR-221-3p, and miR-499a-5p) showed the highest accuracy for prediction of diabetes/cardiovascular risk (96.49% children at 10.0% FPR) in children with abnormal clinical findings with a prior exposure to GDM (AUC 0.975, *p* < 0.001, sensitivity 92.98%, specificity 94.0%, cut off *>0.51799596*) (Figure 4).

### 3.7. Information on microRNA-Target Interactions on Disease Ontologies, Human Phenotype Ontologies, and OMIM Disorders

The extensive file of predicted targets of all microRNAs with altered expression in whole peripheral blood of children descending from GDM complicated pregnancies indicates that a large group of these genes is involved in ontologies of human diseases (obesity, hypertension, glucose intolerance, lipid metabolism disease, type 2 diabetes mellitus, heart septal defects and heart valve disease, heart disease, heart failure, venous insufficiency, and pulmonary embolism) reported by several previous population-based cohort studies [5,6,7,8,9,10,11,12,13,14,15] (Appendix A).

## 4. Discussion

A diabetes/cardiovascular/cerebrovascular disease associated microRNA expression profile was assessed in whole peripheral blood of children at the age of 3 to 11 years with a prior exposure to GDM with the aim to assess to what extent fetal and environmental programming predisposes the affected children to later development of diabetes mellitus, cardiovascular, and cerebrovascular diseases.

The hypothesis of the assessment of potential diabetes/cardiovascular risk in children prenatally exposed to GDM was based on the knowledge that a serious of microRNAs play a role in the pathogenesis of diabetes mellitus and cardiovascular/cerebrovascular diseases (Table 1) [46,47,48,49,50,51,52,53,54,55,56,57,58,59,60,61,62,63,64,65,66,67,68,69,70,71,72,73,74,75,76,77,78,79,80,81,82,83,84,85,86,87,88,89,90,91,92,93,94,95,96,97,98,99,100,101,102,103,104,105,106,107,108,109,110,111,112,113,114,115,116,117,118,119,120,121,122,123,124,125,126,127,128,129,130,131,132,133,134,135,136,137,138,139,140,141,142,143,144,145,146,147,148,149,150,151,152,153,154,155,156,157,158,159,160,161,162,163,164,165,166,167,168,169,170,171,172,173,174,175,176,177,178,179,180,181,182,183,184,185,186,187,188,189,190,191,192,193,194,195,196,197,198,199,200,201,202,203,204,205,206,207,208,209,210,211,212,213,214,215,216,217,218,219,220,221].

Surprisingly, a substantially altered expression profile of diabetes/cardiovascular/cerebrovascular disease associated microRNAs (23/29 studied microRNAs: miR-1-3p, miR-16-5p, miR-17-5p, miR-20a-5p, miR-20b-5p, miR-21-5p, miR-26a-5p, miR-29a-3p, miR-92a-3p, miR-103a-3p, miR-125b-5p, miR-126-3p, miR-133a-3p, miR-143-3p, miR-146a-5p, miR-155-5p, miR-181a-5p, miR-195-5p, miR-199a-5p, miR-210-3p, miR-221-3p, miR-499a-5p, and miR-574-3p) was found in children descending from GDM complicated pregnancies when compared with children descending from pregnancies with a normal course of gestation. Almost all microRNAs with the exception of miR-92a-3p, miR-155-5p, and miR-210-3p were upregulated in whole peripheral blood of children affected with GDM.

Screening based on the combination of microRNAs with a good sensitivity only in the ROC curve analysis was superior over using individual microRNAs for the prediction of potential risk of later development of diabetes mellitus and/or cardiovascular/cerebrovascular diseases, since it was able to identify 88.14% children with an aberrant microRNA expression profile at 10.0% FPR.

Subsequently, we performed analyses to see how the microRNA expression profile differed in relation to the current absence or presence of cardiovascular risk factors and cardiovascular complications (overweight/obesity, prehypertension/hypertension, and/or valve problems and heart defects) and simultaneously to the previous occurrence of maternal pregnancy complication (GDM).

Again, a set of microRNAs associated with diabetes/cardiovascular/cerebrovascular diseases (miR-1-3p, miR-17-5p, miR-20a-5p, miR-20b-5p, miR-21-5p, miR-29a-3p, miR-103a-3p, miR-126-3p, miR-133a-3p, miR-143-3p, miR-181a-5p, miR-195-5p, miR-210-3p, miR-221-3p, and miR-499a-5p) was dysregulated in both groups of children with a prior exposure to GDM regardless of the occurrence of postnatal clinical findings. In addition, seven microRNAs (miR-16-5p, miR-23a-3p, miR-26a-5p, miR-100-5p, miR-125b-5p, miR-146a-5p, and miR-574-3p) were dysregulated in children affected with GDM with normal postnatal clinical findings. An aberrant expression of two additional microRNAs, miR-92a-3p and miR-155-5p, was observed in children with a prior exposure to GDM, who were found to have abnormal clinical findings.

As expected, screening based on the combination of microRNAs with a good sensitivity in the ROC curve analysis only was superior over using individual microRNAs in the assessment of potential risk of later development of diabetes mellitus and cardiovascular/cerebrovascular diseases in groups of children descending from GDM pregnancies with either normal or abnormal clinical findings. At 10.0% FPR, it was able to identify 75.41% and 96.49% children with an aberrant microRNA expression profile.

Subsequently, to assess the impact of cardiovascular risk factors and cardiovascular complications (overweight/obesity, prehypertension/hypertension, and/or valve problems and heart defects) on the microRNA expression profile we further compared the microRNA gene expression profile between equal groups of children descending from pregnancies that had a normal or abnormal course of gestation.

Similarly, as in our previous study [226] dedicated to cardiovascular risk assessment in children descending from pregnancies affected with gestational hypertension, preeclampsia and/or fetal growth restriction, the expression profile of microRNAs was equal between the groups of children descending from uncomplicated pregnancies with normal and abnormal clinical findings, with an exception of miR-21-5p, which showed a trend towards upregulation in a proportion of children with abnormal clinical findings (17.14%).

Parallel, the presence of cardiovascular risk factors (overweight/obesity and/or prehypertension/hypertension) and cardiovascular complications (valve problems and heart defects) had little impact on the microRNA expression profile of children with a prior exposure to GDM, since only miR-29a-3p (12.28%) and miR-92a-3p (12.28%) differentiated between children with normal and abnormal clinical findings (BMI, blood pressure, and echocardiogram findings) with a poor sensitivity.

Previously, we demonstrated that a proportion of children affected with pregnancy-related complications such as gestational hypertension (GH), preeclampsia (PE), and/or fetal growth restriction (FGR) had alterations in the microRNA expression profile that may predispose these children to later development of cardiovascular/cerebrovascular diseases [226,232,233].

Likewise, as a previous occurrence of GH, PE and/or FGR as well as a previous occurrence of GDM was associated with the dysregulation of miR-1-3p, miR-17-5p, miR-20a-5p, miR-20b-5p, miR-21-5p, miR-23a-3p, miR-26a-5p, miR-29a-3p, miR-103a-3p, miR-125b-5p, miR-126-3p, miR-133a-3p, miR-146a-5p, miR-181a-5p, miR-195-5p, miR-210-3p, and miR-342-3p [226,232,233].

On the other hand, our study revealed that dysregulation of miR-16-5p, miR-92a-3p, miR-100-5p, miR-143-3p, miR-155-5p, miR-221-3p, miR-499a-5p, and miR-574-3p represented a unique feature of an aberrant expression profile of children with a prior exposure to GDM. From these findings, it is obvious, that a large proportion of children prenatally exposed to GDM has a substantially altered microRNA expression profile associated with diabetes mellitus and cardiovascular/cerebrovascular diseases.

Generally, epigenetics refers to DNA sequence independent alterations that are responsible for control of transcription. The classic epigenetic modifications include DNA methylation, post-translational modifications of histone proteins, silencing of the extra copy of the X chromosome in women, and genomic imprinting. In addition, proteins and protein complexes with epigenetic modifying capabilities have been involved under the definition of epigenetics. Furthermore, with the discovery of the RNA interference machinery, several classes of noncoding RNAs (microRNAs, small-interfering RNAs, and long noncoding RNAs) have been added to the definition of epigenetics [234]. Nevertheless, controversy still exists as to whether or not microRNAs should be considered as part of the epigenetic program [234,235,236]. While classical epigenetic mechanisms, such as histone modification and DNA methylation, regulate expression at the transcriptional level, microRNAs putatively function mainly at the posttranscriptional level [234,235,236].

Epigenetics mechanisms appear to be interconnected on multiple levels. DNA methylation may direct histone methylation or vice versa chromatin remodelling drives DNA methylation [235,237]. RNAi also appears to be interconnected with DNA methylation and histone modifications. In some species, the link between microRNAs and epigenetics is strong as shown in vitro by transfection of synthetic siRNAs. However, it remains to be seen whether endogenous microRNAs or other types of noncoding RNAs can be also linked to epigenetic mechanisms in vivo [235].

It is also well known that epigenetic modifications are heritable, can be stably transmitted through cell divisions, but can also be reset, since they are very sensitive to the environment and health status [238,239,240,241,242,243,244,245,246,247,248,249]. Recently, the microRNA expression patterns in placental [241,242,244] and germ cells [243,245] have been implicated in fetal programming, and increasing evidence is considering the function of microRNAs in mediating transgenerational epigenetic inheritance [244,246,247,248]. Notably, microRNAs control de novo DNA methylation by regulating transcriptional repressors during germ cell reprogramming [238]. Conversely, global suppression of microRNAs has been observed in mature oocytes and during early embryonic development [239,240]. Consistent with these data, oocytes lack DGCR8 (Pasha), which is necessary for microRNA pathways [239]. It was also demonstrated that many environmental factors contribute to the variations in the epigenome, but diet and early life experiences are key modulators of epigenome, which may initiate the development of the disease [234,249,250,251,252,253,254,255,256,257].

From our findings, it is obvious, that a large proportion of children prenatally exposed to GDM has substantially altered the microRNA expression profile associated with diabetes mellitus and cardiovascular/cerebrovascular diseases, which may be one of several possible reasons for an increased cardiovascular risk. In general, children with a prior exposure to GDM are at higher risk of later development of diabetes mellitus and cardiovascular/cerebrovascular diseases and would benefit from dispensarisation and implementation of primary prevention strategies.

The identification of valve problems and heart defects within the group of children descending from GDM complicated pregnancies more often than usual may support the suggestion for their dispensarisation and implementation of primary prevention strategies.

Similarly, as other studies [258,259,260] we observed that infertility and an infertility treatment may be associated with an increased risk of GDM onset. The incidence of infertility and infertility treatment was significantly higher in groups of mothers with a GDM complicated course of gestation when compared to the group of mothers with normal pregnancies (Table 2). In addition, GDM occurring after assisted reproductive technology conception had been reported to increase the risk of adverse obstetric and perinatal outcomes [259]. This finding may strengthen the idea of dispensarisation of children descending from GDM complicated pregnancies and implementation of primary prevention strategies in this risky group. With regard to a low number of children descending from GDM complicated pregnancies after an infertility treatment (n = 16), it is misleading to interpret the association between microRNA gene expression in whole peripheral venous blood of children and infertility treatment in mothers. Most of the significant results was achieved when the comparison between children descending from normal pregnancies and children descending from GDM complicated pregnancies regardless of infertility treatment was performed or when the comparison between the groups of children descending from normal and GDM complicated pregnancies without infertility treatment was made (83 NP and 102 GDM).

## 5. Conclusions

In conclusion, any of the tissue-specific and circulation-specific (plasma and/or serum) changes in the microRNA expression profile characteristic for patients with diabetes mellitus and cardiovascular/cerebrovascular diseases are also present in whole peripheral blood (white blood cells) of children previously exposed to GDM. This finding indicates that a previous occurrence of GDM may predispose affected children to later development of diabetes mellitus and cardiovascular/cerebrovascular diseases. Consecutive large-scale studies are needed to verify the findings resulting from this pilot study.

## 6. Patents

National patent granted—Industrial Property Office, Czech Republic (Patent no. 308102).

International patent filed—Industrial Property Office, Czech Republic (PCT/CZ2019/050050).

## Figures and Tables

**Figure 1 cells-09-01557-f001:**
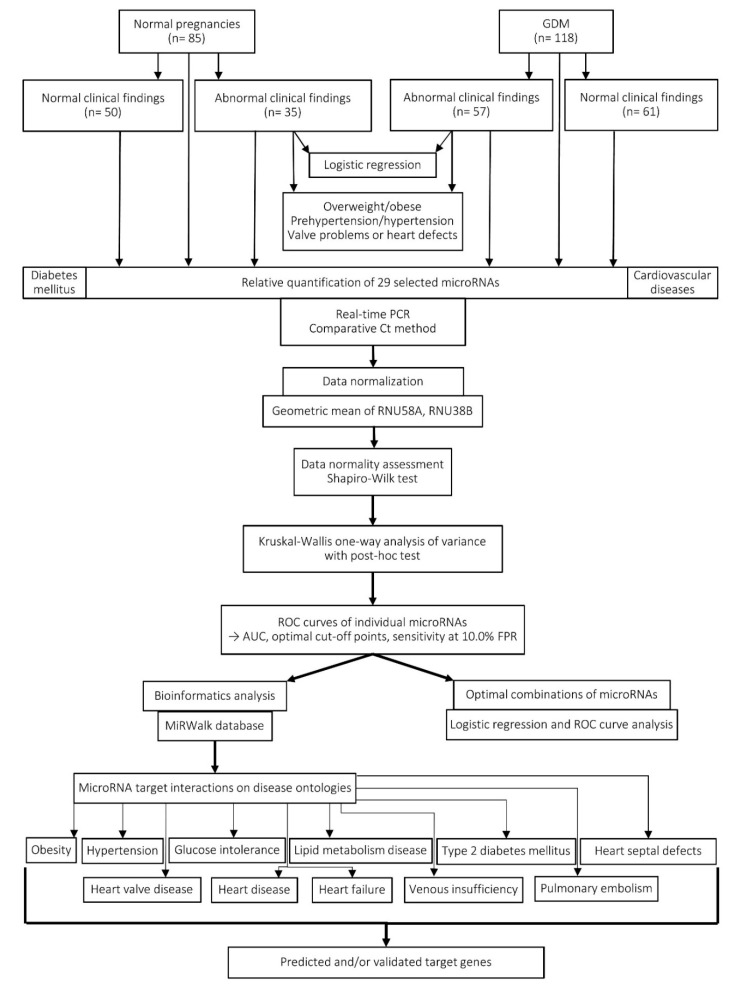
Workflow of the study.

**Figure 2 cells-09-01557-f002:**
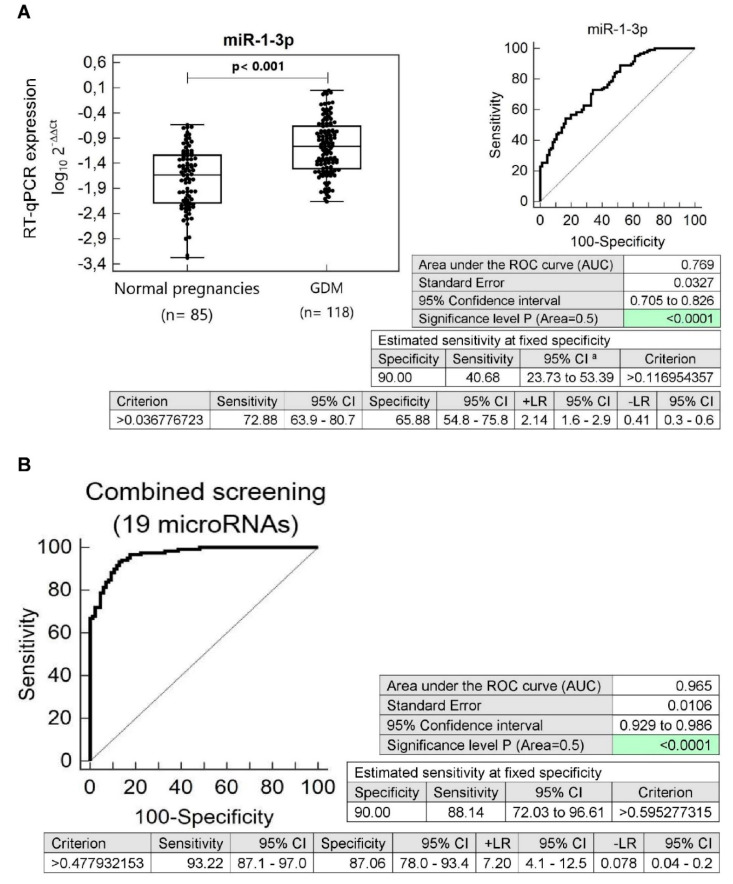
Aberrant microRNA expression profile in children descending from GDM complicated pregnancies irrespective of the clinical findings (overweight/obesity, prehypertension/hypertension, and/or valve problems and heart defects). (**A**) Upregulation of miR-1-3p was observed in children descending from GDM complicated pregnancies when the comparison to the controls irrespective of the clinical findings was performed. Concerning individual microRNAs, miR-1-3p showed the highest accuracy for the identification of children at a higher risk of later development of diabetes mellitus and/or cardiovascular/cerebrovascular diseases. (**B**) Combined screening of microRNAs in the identification of children prenatally exposed to GDM at an increased risk of later development of diabetes mellitus and/or cardiovascular/cerebrovascular diseases. Screening based on the combination of microRNAs with a good sensitivity (miR-1-3p, miR-16-5p, miR-17-5p, miR-20a-5p, miR-20b-5p, miR-21-5p, miR-26a-5p, miR-29a-3p, miR-103a-3p, miR-125b-5p, miR-126-3p, miR-133a-3p, miR-143-3p, miR-181a-5p, miR-195-5p, miR-210-3p, miR-221-3p, miR-499a-5p, and miR-574-3p) showed the highest accuracy for the identification of children at a higher risk of later development of diabetes mellitus and/or cardiovascular/cerebrovascular diseases. GDM: Gestational diabetes mellitus.

**Figure 3 cells-09-01557-f003:**
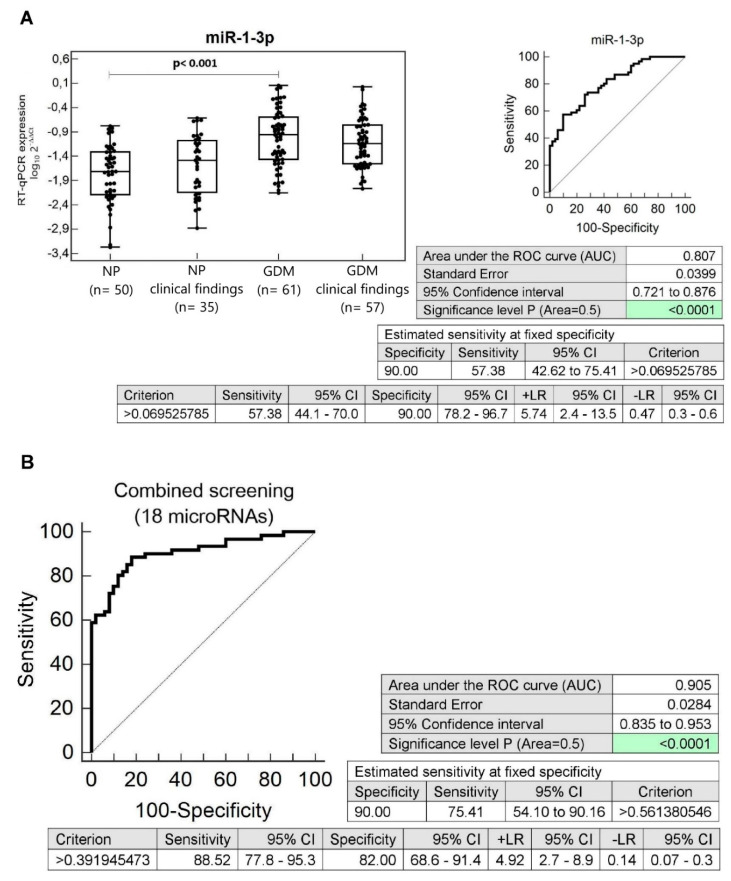
Aberrant microRNA expression profile in children descending from GDM complicated pregnancies with normal clinical findings. (**A**) Upregulation of miR-1-3p was observed in children descending from GDM complicated pregnancies with normal clinical findings, when the comparison to the controls with normal clinical findings was performed. Concerning individual microRNAs, miR-1-3p showed the highest accuracy for the identification of children at a higher risk of later development of diabetes mellitus and/or cardiovascular/cerebrovascular diseases. (**B**) Combined screening of microRNAs in the identification of children prenatally exposed to GDM with normal clinical findings at an increased risk of later development of diabetes mellitus and/or cardiovascular/cerebrovascular diseases. Screening based on the combination of microRNAs with a good sensitivity (miR-1-3p, miR-17-5p, miR-20a-5p, miR-20b-5p, miR-21-5p, miR-23a-3p, miR-29a-3p, miR-100-5p, miR-103a-3p, miR-125b-5p, miR-126-3p, miR-133a-3p, miR-143-3p, miR-181a-5p, miR-195-5p, miR-221-3p, miR-499a-5p, and miR-574-3p) showed the highest accuracy for the identification of children prenatally exposed to GDM with normal clinical findings at a higher risk of later development of diabetes mellitus and/or cardiovascular/cerebrovascular diseases. The comparison to the controls with normal clinical findings was performed. NP: Normal pregnancies; GDM: Gestational diabetes mellitus.

**Figure 4 cells-09-01557-f004:**
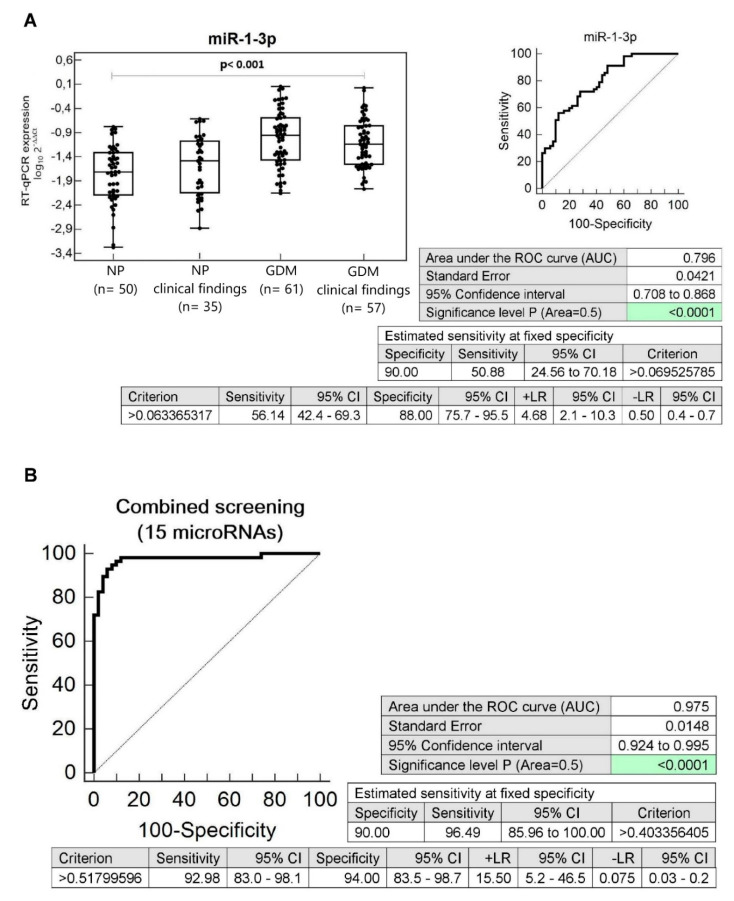
Aberrant microRNA expression profile in children descending from GDM complicated pregnancies with abnormal clinical findings. (**A**) Upregulation of miR-1-3p was observed in children descending from GDM complicated pregnancies with abnormal clinical findings, when the comparison to the controls with normal clinical findings was performed. Concerning individual microRNAs, miR-1-3p showed the highest accuracy for the identification of children at a higher risk of later development of diabetes mellitus and/or cardiovascular/cerebrovascular diseases. (**B**) Combined screening of microRNAs in the identification of children prenatally exposed to GDM with abnormal clinical findings at an increased risk of later development of diabetes mellitus and/or cardiovascular/cerebrovascular diseases. Screening based on the combination of microRNAs with a good sensitivity (miR-1-3p, miR-17-5p, miR-20a-5p, miR-20b-5p, miR-21-5p, miR-29a-3p, miR-92a-3p, miR-103a-3p, miR-126-3p, miR-133a-3p, miR-143-3p, miR-181a-5p, miR-195-5p, miR-221-3p, and miR-499a-5p) showed the highest accuracy for the identification of children prenatally exposed to GDM with abnormal clinical findings at a higher risk of later development of diabetes mellitus and/or cardiovascular/cerebrovascular diseases. The comparison to the controls with normal clinical findings was performed. NP: Normal pregnancies; GDM: Gestational diabetes mellitus.

**Table 1 cells-09-01557-t001:** The role of studied microRNAs in the pathogenesis of diabetes mellitus and cardiovascular/cerebrovascular diseases.

miRBase ID	Gene Location on Chromosome	Role in the Pathogenesis of Diabetes Mellitus and Cardiovascular/Cerebrovascular Diseases
hsa-miR-1-3p	20q13.3 [46] 18q11.2	Acute myocardial infarction, heart ischemia, post-myocardial infarction complications [47], thoracic aortic aneurysm [48], diabetes mellitus [49,50], vascular endothelial dysfunction [51].
hsa-miR-16-5p	13q14.2	Myocardial infarction [52,53], heart failure [54], acute coronary syndrome, cerebral ischaemic events [55], gestational diabetes mellitus [9,13,20], diabetes mellitus [56,57,58].
hsa-miR-17-5p	13q31.3 [59,60]	Cardiac development [61], ischemia/reperfusion-induced cardiac injury [62], kidney ischemia-reperfusion injury [63], diffuse myocardial fibrosis in hypertrophic cardiomyopathy [64], acute ischemic stroke [65], coronary artery disease [66], adipogenic differentiation [67], gestational diabetes mellitus [9,13], diabetes mellitus [58,68].
hsa-miR-20a-5p	13q31.3 [69]	Pulmonary hypertension [70], gestational diabetes mellitus [9,13,17], diabetic retinopathy [71], diabetes with abdominal aortic aneurysm [72].
hsa-miR-20b-5p	Xq26.2 [69]	Hypertension-induced heart failure [73], insulin resistance [74], T2DM [75,76], diabetic retinopathy [77].
hsa-miR-21-5p	17q23.2 [78]	Homeostasis of the cardiovascular system [79], cardiac fibrosis and heart failure [80,81], thoracic aortic aneurysm [48], ascending aortic aneurysm [82], regulation of hypertension-related genes [83], myocardial infarction [84], insulin resistance [74], T2DM [85], T2DM with major cardiovascular events [86], T1DM [87,88,89], diabetic nephropathy [90].
hsa-miR-23a-3p	19p13.12	Heart failure [91], coronary artery disease [92], cerebral ischemia-reperfusion [93], vascular endothelial dysfunction [51], small and large abdominal aortic aneurysm [94], obesity and insulin resistance [95].
hsa-miR-24-3p	19p13.12	Asymptomatic carotid stenosis [96], familial hypercholesterolemia and coronary artery disease [97], angina pectoris [98], ischemic dilated cardiomyopathy [99], small and large abdominal aortic aneurysm [94], myocardial ischemia/reperfusion [100,101], diabetes mellitus [50,58,62,64].
hsa-miR-26a-5p	3p22.2 [102] 12q14.1	Heart failure, cardiac hypertrophy [103], myocardial infarction [84,104,105], ischemia/reperfusion injury [106], pulmonary arterial hypertension [107], T1DM [108], diabetic nephropathy [90].
hsa-miR-29a-3p	7q32.3	Ischemia/reperfusion-induced cardiac injury [109], cardiac cachexia, heart failure [110], atrial fibrillation [111], diffuse myocardial fibrosis in hypertrophic cardiomyopathy [64], coronary artery disease [112], pulmonary arterial hypertension [107], gestational diabetes mellitus [12], diabetes mellitus [49,57,113,114].
hsa-miR-92a-3p	13q31.3 Xq26.2	Mitral chordae tendineae rupture [115], children with rheumatic carditis [116], myocardial infarction [117], heart failure [118], coronary artery disease [119], renal injury-associated atherosclerosis [120].
hsa-miR-100-5p	11q24.1	Failing human heart, idiopathic dilated cardiomyopathy, ischemic cardiomyopathy [99], regulation of hypertension-related genes [83], T1DM [87].
hsa-miR-103a-3p	5q34 [121] 20p13	Hypertension [122], hypoxia-induced pulmonary hypertension [123], myocardial ischemia/reperfusion injury, acute myocardial infarction [124], ischemic dilated cardiomyopathy [99], obesity, regulation of insulin sensitivity [125], T1DM [126].
hsa-miR-125b-5p	11q24.1 [126] 21q21.1	Acute ischemic stroke [127], acute myocardial infarction [128,129], ischemic dilated cardiomyopathy [99], ascending aortic aneurysm [82], gestational diabetes mellitus [19], T1DM [130,131], T2DM [132].
hsa-miR-126-3p	9q34.3 [133]	Acute myocardial infarction [105], thoracic aortic aneurysm [48], T2DM [86,134], T2DM with major cardiovascular events [86], gestational diabetes mellitus [10].
hsa-miR-130b-3p	22q11.21	Hypertriglyceridemia [135,136], intracranial aneurysms [137], hyperacute cerebral infarction [138], T2DM [85,139,140], gestational diabetes mellitus [10].
hsa-miR-133a-3p	18q11.2 [141] 20q13.33	Heart failure [142], myocardial fibrosis in hypertrophic cardiomyopathy [64,143], arrhythmogenesis in the hypertrophic and failing hearts [144,145], coronary artery calcification [146], thoracic aortic aneurysm [48], ascending aortic aneurysm [82], diabetes mellitus [49,50].
hsa-miR-143-3p	5q33	Intracranial aneurysms [147], coronary heart disease [148], myocardial infarction [149], myocardial hypertrophy [150], dilated cardiomyopathy [151], pulmonary arterial hypertension [152], acute ischemic stroke [127], ascending aortic aneurysm [82].
hsa-miR-145-5p	5q33	Hypertension [153,154], dilated cardiomyopathy [155], myocardial infarction [156,157], stroke [157], acute cerebral ischemic/reperfusion [158], T2DM [58,159], T1DM [85], diabetic retinopathy [160], gestational diabetes mellitus [161].
hsa-miR-146a-5p	5q33.3 [162,163]	Angiogenesis [164], hypoxia, ischemia/reperfusion-induced cardiac injury [165], myocardial infarction [53], coronary atherosclerosis, coronary heart disease in patients with subclinical hypothyroidism [166], thoracic aortic aneurysm [48], acute ischemic stroke, acute cerebral ischemia [167], T2DM [58,85], T1DM [108], diabetic nephropathy [90].
hsa-miR-155-5p	21q21.3	Thoracic aortic aneurysm [48], type 1 diabetes [125], gestational diabetes mellitus [20], adolescent obesity [168], diet-induced obesity and obesity resistance [168], atherosclerosis [170], hyperlipidemia—associated endotoxemia [171], coronary plaque rupture [172], children with cyanotic heart disease [173], chronic kidney disease and nocturnal hypertension [174], atrial fibrillation [175].
hsa-miR-181a-5p	1q32.1 [176] 9q33.3	Regulation of hypertension-related genes [65], atherosclerosis [176], metabolic syndrome, coronary artery disease [177], non-alcoholic fatty liver disease [178], ischaemic stroke, transient ischaemic attack, acute myocardial infarction [179,180], obesity and insulin resistance [95,176,177], T1DM [85,181], T2DM [176,180].
hsa-miR-195-5p	17p13.1 [182]	Cardiac hypertrophy, heart failure [183,184], abdominal aortic aneurysms [185], aortic stenosis [186], T2DM [159], gestational diabetes mellitus [18].
hsa-miR-199a-5p	1q24.3 19p13.2	T1DM, T2DM, gestational diabetes mellitus [187], diabetic retinopathy [188], cerebral ischemic injury [189], heart failure [190], hypertension [191,192], congenital heart disease [193], pulmonary artery hypertension [194], unstable angina [195], hypoxia in myocardium [193], acute kidney injury [196].
hsa-miR-210-3p	11p15.5	Cardiac hypertrophy [197], acute kidney injury [198], myocardial infarction [199], atherosclerosis [200].
hsa-miR-221-3p	Xp11.3	Asymptomatic carotid stenosis [96], cardiac amyloidosis [201], heart failure [202], atherosclerosis [203,204], aortic stenosis [205], acute myocardial infarction [206], acute ischemic stroke [207], focal cerebral ischemia [208], pulmonary artery hypertension [209], obesity [210].
hsa-miR-342-3p	14q32.2	Cardiac amyloidosis [201], obesity [211], T1DM [85,187,212], T2DM [187,213,214], gestational diabetes mellitus [187], endothelial dysfunction [215].
hsa-miR-499a-5p	20q11.22	Myocardial infarction [53,216], hypoxia [217], cardiac regeneration [218], vascular endothelial dysfunction [51].
hsa-miR-574-3p	4p14	Myocardial infarction [219], coronary artery disease [136], cardiac amyloidosis [201], stroke [220], T2DM [140,221].

T1DM: Diabetes mellitus type 1; T2DM: Diabetes mellitus type 2.

**Table 2 cells-09-01557-t002:** Characteristics of cases and controls.

	Normal Pregnancies Normal Clinical Findings (*n* = 48)	Normal Pregnancies Abnormal Clinical Findings (*n* = 37)	Gestational Diabetes Mellitus (GDM) Normal Clinical Findings (*n* = 61)	GDM Abnormal Clinical Findings (*n*= 57)	*p*-Value^1^	*p*-Value^2^	*p*-Value^3^
Children at follow-up							
Age (years)	5 (3–11)	5 (3–11)	5 (3–10)	5 (3–9)	0.228	0.980	0.358
Height (cm)	115.5 (98–144.5)	118.0 (100–153)	114.0 (99–143.5)	113.5 (98–153)	0.332	0.546	0.274
Weight (kg)	20.8 (14–37)	22.3 (14.7–40.8)	19.5 (14.4–37.4)	19.6 (15–47.1)	0.104	0.604	0.565
BMI (kg/m^2^)	15.41 (13.22–18.09)	15.80 (13.3–20)	15.20 (13.53–18.09)	15.58 (12.97–20.08)	0.067	0.976	0.166
Systolic BP (mmHg)	98 (84–115)	104 (89–123)	99 (82–113)	101 (87–125)	<0.001	0.864	0.027
Diastolic BP (mmHg)	60 (38–68)	64.0 (43–81)	60 (47–67)	61 (49–79)	0.002	0.795	0.008
Heart rate (n/min)	90 (67–110)	90.0 (51–120)	96 (64–118)	98 (78–122)	0.954	0.022	<0.001
During Gestation							
Maternal age at delivery (years)	31.5 (21–40)	32 (25–46)	34 (27–42)	33 (27–45)	0.233	0.009	0.046
GA at delivery (weeks)	39.86 (37.71–41.57)	39.86 (37.86–41.86)	39.72 (37.43–41.28)	39.43 (37.00–41.14)	0.852	0.111	0.020
Fetal birth weight (g)	3425 (2730–4220)	3280 (2530–4450)	3500 (2700-4330)	3420 (2770–4400)	0.989	0.157	0.252
Mode of delivery					0.698	<0.001	0.001
Vaginal	44 (91.67%)	33 (89.19%)	38 (62.30%)	37 (64.91%)			
CS	4 (8.33%)	4 (10.81%)	23 (37.70%)	20 (35.09%)			
Fetal sex					0.346	0.771	0.273
Boy	27 (56.25%)	17 (45.95%)	36 (59.02%)	38 (66.67%)			
Girl	21 (43.75%)	20 (54.05%)	25 (40.98%)	19 (33.33%)			
Primiparity					0.069	0.181	0.091
Yes	29 (60.42%)	15 (40.54%)	29 (47.54%)	25 (43.86%)			
No	19 (39.58%)	22 (59.46%)	32 (52.46%)	32 (56.14%)			
Birth order of index pregnancy					0.058	0.080	0.294
1st	26 (54.17%)	11 (29.73%)	20 (32.79%)	22 (38.60%)			
2nd	16 (33.33%)	14 (37.84%)	23 (37.70%)	22 (38.60%)			
3rd	4 (8.33%)	10 (27.03%)	10 (16.39%)	6 (10.52%)			
4th+	2 (4.17%)	2 (5.40%)	8 (13.11%)	7 (12.28%)			
Infertility treatment					0.852	0.014	0.084
Yes	1 (2.08%)	1 (2.70%)	10 (16.39%)	6 (10.53%)			
No	47 (97.92%)	36 (97.30%)	51 (83.61%)	51 (89.47%)			
Maternal BMI (kg/m^2^)							
Prepregnancy BMI	21.88 (14.77–30.3)	21.22 (17.37–28.04)	22.02 (16.3–30.85)	22.64 (17.53–30.49)	1.000	1.000	0.567
BMI < 18.5	4 (8.33%)	5 (13.51%)	5 (8.20%)	1 (1.75%)	-	-	-
BMI 18.5–24.9	38 (79.17%)	31 (83.78%)	43 (70.49%)	37 (64.91%)			
BMI 25.0–29.9	5 (10.42%)	1 (2.70%)	11 (18.03%)	17 (29.82%)			
BMI > 30	1 (2.08%)	0 (0%)	2 (3.28%)	2 (3.51%)			
BMI at admission for delivery	26.17 (20.88–34.82)	26.45 (20.72–33.17)	25.97 (19.84–36.85)	27.53 (20.18–36.73)	1.000	1.000	1.000
Total gestational weight gain (GWG) (kg)	14.5 (8–25.5)	14.5 (8–21)	10 (2–21)	11 (3–26)	1.000	<0.001	0.008
BMI at follow-up	23.26 (17.7–39.08)	22.11 (18.17–29.17)	23.82 (17.39–32.14)	23.42 (17.39–34.37)	0.414	1.000	1.000
Serum Metabolic Biochemical Parameters of Mothers During the Third Trimester of Gestation							
HbA1c	-	-	32 (26–42)	32.5 (26–40)	-	-	-
Creatinine (μmol/L)	56.0 (44.0–70.0)	52.5 (43.0–83.0)	56.0 (40.0–78.0)	55.0 (36.0–83.0)	1.000	1.000	1.000
Uric acid (μmol/L)	286.5 (200.0–377.0)	294.5 (221.0–345.0)	298 (180.0–419.0)	289 (157.0–471.0)	1.000	1.000	1.000
Total bilirubin (μmol/L)	4.0 (3.0–7.0)	4.0 (0.18–12.0)	6.0 (2.70–20.0)	6.0 (3.0–20.0)	1.000	0.497	0.108
ALT (μkat/L)	0.22 (0.11–2.4)	0.17 (0.07–0.36)	0.23 (0.11–1.09)	0.22 (0.11–0.46)	1.000	1.000	1.000
AST (μkat/L)	0.38 (0.25–2.36)	0.47 (0.22–2.29)	0.38 (0.22–1.46)	0.40 (0.24–0.84)	1.000	1.000	1.000
ALP (μkat/L)	2.12 (1.86–4.66)	2.51 (1.71–3.52)	2.67 (1.35–6.15)	2.36 (1.46–5.32)	1.000	1.000	1.000
Cholesterol (mmol/L)	7.50 (6.30–11.40)	9.2 (6.90–10.5))	7.71 (4.47–9.5)	6.93 (6.13–9.10)	1.000	1.000	1.000
Triglyceride (mmol/L)	3.10 (3.00–3.20)	2.9 (2.7–3.3)	3.05 (1.80–4.80)	3.35 (2.50–8.90)	1.000	1.000	1.000
Total protein (g/L)	63.7 (48.7–70.8)	64,35 (47.2–68.9)	60.3 (37.7–70.8)	61.6 (43.8–67.9%)	1.000	1.000	1.000
Albumin (g/L)	37.35 (28.9–40.6)	36.9 (27.9–41.6)	36.5 (4.40–42.1)	36.2(26.3–40.9)	1.000	1.000	1.000
Blood glucose (mmol/L)	4.7 (4.2–5.1)	4.5 (4.3–5.8)	4.65 (4.1–8.0)	4.4 (3.8–5.6)	1.000	1.000	1.000

Data are presented as a median (range) for continuous variables and as a number (percent) for categorical variables. Continuous variables were compared using the non-parametric Kruskal-Wallis test. *P*-value^1^: The comparison among children from the control group with normal and abnormal postnatal clinical findings; *p*-value^2^: The comparison among children descending from normal and GDM complicated pregnancies with normal postnatal clinical findings; *p*-value^3^: The comparison among children descending from normal pregnancies with normal postnatal clinical findings and children descending from GDM complicated pregnancies with abnormal postnatal clinical findings. Categorical variables were compared using a chi-square test. GDM: Gestational diabetes mellitus; BP: Blood pressure; CS: Caesarean section; GA: Gestational age; GWG: Gestational weight gain; HbA1c: Haemoglobin A1c; ALT: Alanine aminotransferase; AST: Aspartate aminotransferase; ALP: Alkaline phosphatase.

**Table 3 cells-09-01557-t003:** The results of clinical examination in children descending from normal and GDM complicated pregnancies.

	Normal Pregnancy	GDM Pregnancy	OR (95% CI)	*p*-Value
Overweight/obese	8/85 (9.41%)	6/118 (5.08%)	0.515 (0.172–1.545)	0.237
Prehypertension/ hypertension	15/85 (17.65%)	19/118 (16.10%)	0.896 (0.426–1.883)	0.771
Valve problems or heart defects	17/85 (20.0%)	42/118 (35.59%)	2.240 (1.167–4.300)	0.015

Logistic regression was used to compare the presence of abnormal clinical findings between particular groups. The significance level was established at a *p*-value of p < 0.05. No difference in the incidence of overweight/obesity and/or prehypertension/hypertension was found between the groups of children descending from normal and GDM complicated pregnancies. A higher incidence of valve problems and heart defects was observed in a group of children descending from GDM complicated pregnancies.

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
