# Peer review of "Substantially Altered Expression Profile of Diabetes/Cardiovascular/Cerebrovascular Disease Associated microRNAs in Children Descending from Pregnancy Complicated by Gestational Diabetes Mellitus—One of Several Possible Reasons for an Increased Cardiovascular Risk"

_cells, 2020, doi:10.3390/cells9061557_

Round 1

Reviewer 1 Report

The manuscript must be revised in a great manner. The data represented in a very bad way, it is hard to read the figures etc. Most of the data should be represented as supplementary materials.. Therefore, at the present time the scientific revision of the present manuscript is worthless and such a waste of time.

I suggest a major revision of manuscript representation (20 page manuscript would be great) before the scientific revision.

Author Response

Response to Reviewer 1 is attached.

Reviewer 2 Report

Evidence demonstrates that abnormal fetal and environmental programming predisposed the affected children to development of metabolic complications via epigenetic modifications. So the aim of this paper was to determine whether exposure to hyperglycemia in utero, as occurs in gestational diabetes mellitus (GDM) induced specific expression profile alteration of several microRNAs associated with metabolic disorders. Thanks to a cohort of children descending from GDM complicated pregnancies, they identified several microRNAs whose expression was specifically altered in GDM children compared to children descending from normal pregnancies Those microRNAs are known to be deregulated in patients suffering from diabetes/cardiovascular/cerebrovascular diseases.

From their data, they conclude that “substantially altered expression profile of

 diabetes/cardiovascular/cerebrovascular disease associated microRNAs in children descending from  pregnancy complicated by gestational diabetes  mellitus – cause of a potential cardiovascular risk”. However, several points should be clarified before reaching this conclusion.

Major revisions:

  • no data on the metabolic parameters of the mothers specifically (BMI). Because of the importance of BMI on the metabolic health of the offspring, this table should be included in the paper.
  • Table explaining the selection of the microRNA. This table called Table 1 in the conclusion is missing in the present manuscript. This table should be included.
  • These microRNAs have been selected thanks to their potential role in diabetes/cardiovascular disease. So it is not very surprising that the targets of these genes are potentially involved in the development of these diseases. In my opinion, this specific result is not sufficient to state that the deregulation of these specific microRNAs is a cause of a potential cardiovascular risk in these children.
  • How the children age affect microRNA expression?
  • How the authors explained the difference between control and GDM in terms of infertility treatment. What about the role of this parameter on the microRNA expression in the corresponding children?
  • Figure 1: this figure is not informative and should be replaced by the mediane of microRNA expression in each group. The information concerning the choice of these specific microRNA should be clearly specified.
  • The figures are quite redundant, in particular the Figures 5 and 6. What is the difference between the left parts of each panel in those figures?
  • The authors employed “aberrant epigenetic profile” on many occasions (lines 24, 493, 511, 514, 524 and 548). They should explain how/why an alteration in microRNA expression can be associated with an abnormal epigenetic profile.
  • The authors should explain how their result are fundamentally different from those recently published in Hromadnikova et al (Hromadnikova, I.; Kotlabova, K.; Dvorakova, L.; Krofta, L.; Sirc, J. Postnatal Expression Profile of microRNAs 1 Associated with Cardiovascular and Cerebrovascular Diseases in Children at the Age of 3 to 11 Years in Relation to Previous Occurrence of Pregnancy-Related Complications. Int J Mol Sci 2019,20, E654).

Author Response

Response to Reviewer 2 is attached.

Reviewer 3 Report

 Comments

  1. Inclusion criteria of cohort    Regarding to IRB guidelines, is there any range of BMI of samples? How long did the trail project last? Please provide start and end timepoint.
  2. Target microRNAs profile      How did authors select target micorRNAs? Please show screening criteria.

     3. Format of article  The current manuscript contains pretty much data, but it's not readable. Please follow authors instructions before submission, and reedit text, figures and tables.

Author Response

Response to Reviewer 3 is attached.

Round 2

Reviewer 1 Report

Major concerns:

1) the title of the manuscript is not clear and clearly understandable: 

"...expression profile of... - cause of..". To my opinion, the profile of something is not a cause of something, it is more as a result of something - please rephrase it. 

2) the abstract starts depicting the aim. Please introduce few sentences about the research object first. What is GDM? - the authors should avoid using abbreviations in the abstract. 

3) the introduction needs a clearer narrative. Please, clearly introduce what is GDM, why GDM and the perspective of miRNA profiling for GDM, leaving the smaller details behind. 3 last paragraphs should be rephrased as one in a clearer way to understand quickly the main problem and the aim of the present research. In addition, many miRNAs (for instance mir-155-5p, mir-126-3p, miR-21 and other) depicted in table 1 were recently associated with some risks of ascending aortic aneurysm (doi.org/10.3390/jcm8101609) - please add this reference into the table and reference list.

4) for the materials - please introduce participant and analysis schema, which should depict the main groups of the participants, number of miRNAs, and the variety of statistical analyses applied in the present manuscript - it would be much clearer to understand. 

minor concerns:

a) please, give the manuscript for experienced english reader. There is much of some stylish errors, which could be improved. The manuscript could sound much stronger after that;

b) 2.7 section could be depicted in introduction;

Author Response

Reply to the Review 1 is attached.

Reviewer 2 Report

I would like to thank the authors for their response to my comments. I found the revised manuscript has improved significantly although several points should be completed before publication (see in the attached document my comments in green).

Author Response

Reply to the Review 2 is attached.

Reviewer 3 Report

Comments:

1:Table 1 just lays out functional of representative microRNAs, and can be put into supplemental part.

2. Validation of representative miRNAs exspression

Please provide data on fold changes of key miRNAs expression through dot blot chart.

Author Response

Reply to the Review 3 is attached.

Round 3

Reviewer 1 Report

The authors answered all my questions.